

# Earth system modeling on Modular Supercomputing Architectures: coupled atmosphere-ocean simulations with ICON 2.6.6-rc

Abhiraj Bishnoi[1, 5], Olaf Stein[1], Catrin I. Meyer[1], René Redler[2], Norbert Eicker[1,3], Helmuth Haak[2], Lars Hoffmann[1], Daniel Klocke[2], Luis Kornblueh[2], and Estela Suarez[1,4]

[1]Jülich Supercomputing Centre, Forschungszentrum Jülich, Jülich, Germany
[2]Max-Planck-Institut für Meteorologie, Hamburg, Germany
[3]School of Mathematics and Natural Sciences, University of Wuppertal, Wuppertal, Germany
[4]Institute of Computer Science, University of Bonn, Bonn, Germany
[5]Kalkulo AS, Oslo, Norway

**Correspondence:** Abhiraj Bishnoi (abhirajbishnoi@yahoo.in), Olaf Stein (o.stein@fz-juelich.de)

**Abstract.**

The confrontation of complex Earth System model (ESM) codes with novel supercomputing architectures poses challenges to efficient modelling and job submission strategies. The modular setup of these models naturally fits a modular supercomputing architecture (MSA), which tightly integrates heterogeneous hardware resources into a larger and more flexible high performance computing (HPC) system. While parts of the ESM codes can easily take advantage of the increased parallelism and communication capabilities of modern Graphics Processing Units (GPUs), others lack behind due to the long development cycles or are better suited to run on classical CPUs due to their communication and memory usage patterns. To better cope with these imbalances between the development of the model components, we performed benchmark campaigns on the Jülich Wizard for European Leadership Science (JUWELS) modular HPC system. We enabled the weather and climate model ICOsahedral Nonhydrostatic (ICON) to run in a coupled atmosphere-ocean setup, where the ocean and the model I/O is running on the CPU Cluster, while the atmosphere is simulated simultaneously on the GPUs of JUWELS Booster (ICON-MSA). Both, atmosphere and ocean, are running globally with a resolution of 5 km. In our test case, an optimal configuration in terms of model performance (core hours per simulation day) was found for the combination 84 GPU nodes on the JUWELS Booster module and 80 CPU nodes on the JUWELS Cluster module, of which 63 nodes were used for the ocean simulation and the remaining 17 nodes were reserved for I/O. With this configuration the waiting times of the coupler were minimized. Compared to a simulation performed on CPUs only, the MSA approach reduces energy consumption by 59 % with comparable runtimes. ICON-MSA is able to scale up to a significant portion of the JUWELS system, making best use of the available computing resources. A maximum throughput of 170 simulation days per day (SDPD) was achieved when running ICON on 335 JUWELS Booster nodes and 268 Cluster nodes.



## 1 Introduction

The Earth's climate is a highly complicated, dynamic system built out of interacting sub-systems that have distinctly different governing equations and constraints. Now, more than ever before, it is crucial to attain a deeper understanding of the underlying processes that govern the weather and climate of our planet, as well as predict the effect of human activities on its environment (Bony et al., 2015; Ummenhofer and Meehl, 2017). These are the major driving forces behind the Destination Earth initiative of the European Union to create a digital twin of our planet (Bauer et al., 2021b).

Since the dawn of supercomputing in the 1960s, numerical modeling of weather and climate has been one of the most important applications (Manabe and Bryan, 1969; Trenberth and Trenberth, 1992; Houghton, 1996; Flato, 2011). With ever progressing advances in compute technologies, climate models have increased their resolution, complexity, ensemble size, number of scenarios, and simulation length to make use of more powerful computers (Randall et al., 2018). But to optimally exploit new technologies, climate codes need to change or adapt (Bauer et al., 2021a). This is a challenge for complex Earth system models, which must rely on scientific and technical innovation efforts of large communities working on single model components and result in development cycles with different speeds.

The code of more advanced complex climate models is composed of many different kinds of operations, e.g., conditional branch selections, index based indirect addressing, large table access, frequent time based output, and time dependent updating of boundary conditions and implements optimisations adapted to specific computing hardware (Lawrence et al., 2018). These bring features such as blocking, with the capability of changing the innermost loop length at runtime with stride one access. The outermost loop length can be defined as the number of blocks corresponding to the length of the innermost loop, allowing for blocking e.g., with OpenMP. Including additional model components to more accurately represent the Earth's physical system further increases the computational demand. As a result, increasing the processing power does not necessarily translate into faster computations done in single simulations. In concrete terms: since 1990, when the first assessment report of the Intergovernmental Panel for Climate Change (IPCC) was released, the speed of the fastest computer in the world increased by a factor of 1 million from under 1 TFLOPS (1993) to over 1 EFLOPS (2023). In the same time span, climate models refined their horizontal resolution from about 500 km (1990) to about 125 km (2021) - a factor of *only* 128 in computations, considering the two horizontal directions, the time-step length, and doubling the number of vertical levels.

A second pathway to utilize the increased computing power of high performance computing (HPC) systems are storm-resolving class models (Stevens et al., 2020) with resolutions from 5 km (2018) up to 1.25 km (2022). This model resolution marks a breakthrough in climate modelling because important processes, which shape the climate of ocean and atmosphere, can be simulated explicitly. At the same time, relevant aspects of climate change, like weather related hazards, are inherent to the simulations and their statistics can be assessed on a regional scale.

Latest generations of HPC systems consist largely of GPUs as accelerators, which require adapting climate codes to exploit their enhanced computing capabilities (Jiang et al., 2019; Schär et al., 2020; Giorgetta et al., 2022). These new technologies can provide better throughput for climate integrations at lower energy consumption, enabling a new realm of climate modeling in which the climate system can be simulated at the km-scale.



The ICOsahedral Nonhydrostatic (ICON) Weather and Climate Model, which consists of component models for atmosphere, land and ocean, originated as a joint project of the German Weather Service (DWD) and the Max Planck Institute for Meteorology (MPI-M) (Zängl et al., 2015). It has then expanded to involve a growing list of partners, now including the German Climate Computing Center Hamburg, the Centre for Climate System Modeling at ETH Zürich, and the Karlsruhe Institute of Technology. Recently, the model entered the stage of km-scale resolutions to resolve important processes of the climate, like convective storms, ocean eddies, land heterogeneity, and sea ice-leads (Hohenegger et al., 2023; Korn et al., 2022). While the atmosphere component has been adapted and optimized to run on GPUs by means of OpenACC directives (Giorgetta et al., 2022), using the already available inner most loop length adaptation, the performance of the ocean component on CPUs is still satisfactory. This situation presents an ideal scenario to exploit the Modular Supercomputing Architecture (MSA) (Suarez et al., 2019) to improve the efficiency of running such models on large supercomputing systems, in terms of runtime for the model itself, as well as resource efficiency of the supercomputing system.

The Modular Supercomputing Architecture concept is a novel way of organizing computing resources in HPC systems, developed at the Jülich Supercomputing Centre (JSC) in the course of the DEEP projects (Suarez et al., 2018). Its key idea is to segregate hardware resources (e.g. CPUs, GPUs, other accelerators) into separate modules such that the nodes within each module are maximally homogeneous. This approach has the potential to bring substantial benefits for heterogeneous applications such as ICON, wherein each sub-component has been tailored or is by nature of the numerical solution better suited to run efficiently on specific hardware, corresponding to differences in the governing physics. Another intriguing target for running on target specific hardware is using asynchronous output strategies based on remote memory access. Optimizing the network utilization for such a usage model supports performance objectives by reducing network jitter due to different network usage patterns for the components. In such applications, each part can ideally be run on an exactly matching module of the MSA, improving time to solution and energy efficiency.

The primary focus of this work is to explore and empirically describe the benefits of using the MSA for running large scale climate simulations, using ICON coupled atmosphere-ocean simulations as an exemplary scientific use case. A second focus is to describe the process and challenges associated with undertaking such an endeavour. The ICON model was ported to and optimized on the MSA-system Jülich Wizard for European Leadership Science (JUWELS) Jülich Supercomputing Centre (2019), in such a manner that the atmosphere component of ICON is run on GPU equipped nodes, while the ocean component runs on standard CPU-only nodes. The output servers, responsible for performing I/O, also run on CPU-only nodes. This setup allows for an optimal use of the GPU nodes by fully dedicating not only the GPU usage but as well the per GPU available network adapters.

In the next section we provide a comprehensive description of the ICON model and its specific setup. Section 3 presents a brief overview of the MSA, starting with an introduction to the concept. Section 3.2 presents the modular hardware and software architecture of the JUWELS system at JSC. Next, in Sect. 3.3, the strategy for porting the ICON model to the MSA is described, with a detailed explanation of the rationale behind each decision we made. Results from our analyses for finding a sweet spot configuration for ICON, the comparison to a non-modular setup, and strong scaling tests are provided in Sect. 4.





Specific challenges and considerations associated with porting such complex codes as ICON to the MSA are discussed in Sect. 5. Section 6 provides the summary and conclusions of this study.

## 2 Model description

The global coupled ICON model setup used in this study (Fig. 1) is based on the non-hydrostatic atmosphere employing the Sapphire physics (Hohenegger et al., 2023), furtheron referred to as ICON-A, and a hydrostatic ocean including sea ice (ICON-O) (Korn et al., 2022). The land surface model ICON-LAND is integral part of the ICON-A code and its configuration will not be discussed here any further. The setup used here is described in detail as the G_AO_5km setup by Hohenegger et al. (2023). Here we repeat only those characteristics of the model components that are key for the technical focus of this paper. The horizontal grid resolution is 5 km in both components, land cells in the ocean component are excluded. Thus, the grids encompass 20,971,520 cells for the atmosphere and 14,886,338 for the ocean. In the vertical, the ocean is discretized on 128 z-levels, in the atmosphere a terrain following hybrid sigma z-coordinate (the Smooth LEvel VErtical coordinate (SLEVE), Leuenberger et al., 2010) is used with 90 vertical levels. Compared to coarser resolution climate models this configuration allows for a better representation of atmospheric convection and mesoscale eddies in the ocean (Schär et al., 2020).

Atmosphere and ocean interact tightly, exchanging surface fluxes at the sea surface interface. In this interaction, the atmosphere model provides zonal and meridional components of the wind stress, surface freshwater flux (rain, snow, evaporation, river discharge), short- and long-wave radiation, as well as latent and sensible heat fluxes, sea ice surface and bottom melt potentials, 10 m wind speed and sea level pressure. The ocean provides sea surface temperature, zonal and meridional components of velocity at the sea surface, ice and snow thickness, as well as ice concentration. Technically the data exchange between the two components is implemented via Yet Another Coupler (YAC, Hanke et al., 2016), here using a more recent version 2.6.1. The geographical positions of ocean cell centers and vertices are identical in both components. Therefore, no interpolation is required and we use the 1-nearest-neighbour search to repartition all coupling fields but the river discharge. The latter is transferred via source-to-target mapping where the discharge, which is provided on single coastal cells in the atmosphere domain, is remapped to a wider range of coastal ocean cells.

While it is believed that many physical processes are already represented on kilometer scales within the non-hydrostatic equations of motion and conservation laws, which form the basis of ICON (Zängl et al., 2015), a few others still need to be parameterized due to their small-scale, especially radiation, microphysics, and turbulence. The parametrizations for these processes and their calling sequences are described in detail in Hohenegger et al. (2023).

### 2.1 Model input data

For our experiments, ICON-A is initialized using ECMWF's Integrated Forecasting System (IFS) operational analysis data on 20 January 2020, 0 UTC. The ocean initial state for the same date is taken from an uncoupled ICON ocean model simulation with ERA-5 reanalysis forcing Hersbach et al. (2020). Details of the ocean spinup can be found in Hohenegger et al. (2023). Several external parameters need to be provided as boundary or background conditions. This includes orography and



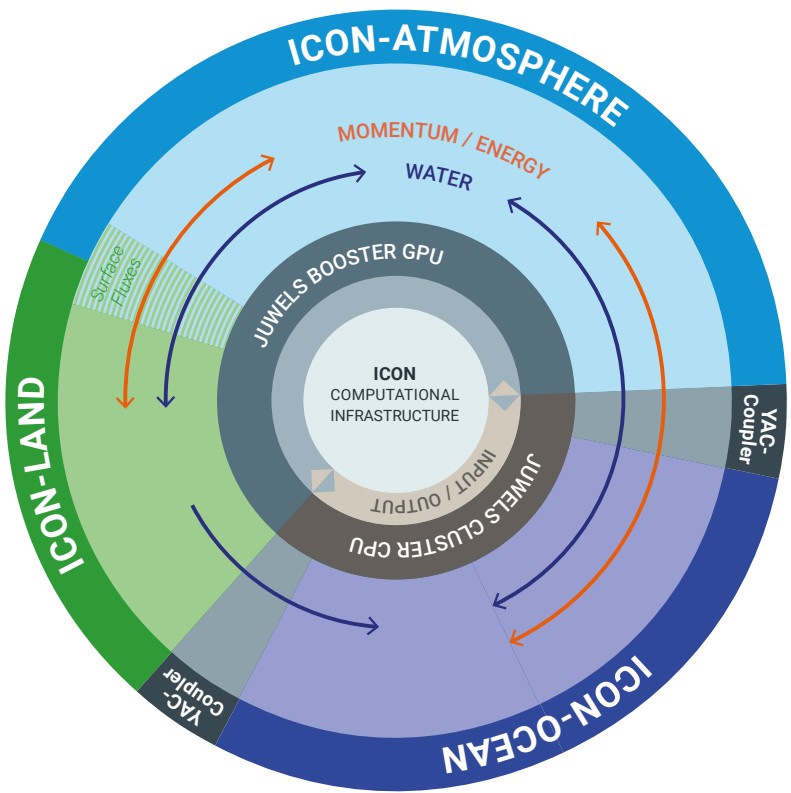

**Figure 1.** The ICON Earth system model, its components and exchange infrastructures.

bathymetry, as well as global mean concentrations of greenhouse gases and monthly varying fields of ozone concentrations and aerosols. The data sets used here are described in more detail in Hohenegger et al. (2023). Our simulations are running with a horizontal grid spacing of 5 km (R2B9). Input data for these simulations add up to a total of 793 GByte and are provided as files with global coverage in NetCDF format following the CF-conventions (Hassell et al., 2017), which are already interpolated onto the horizontal model grid. As an exception, the ocean initialization is performed in parallel from multiple patch files.

## 2.2 Model output data

To allow for long-time (in the model's world) integrations, one single program run is a bad choice. The reason is that it can never be assured that the underlying compute system is sufficiently stable, which forces the requirement for the model integration to be able to checkpoint/restart. Furthermore, most sites limit the runtime of a single job to several hours. E.g., for the systems in Jülich single jobs are limited to 24 hours. System checkpoint/restart of big model codes requires large amounts of data to be dumped to disk. To minimize this, ICON uses a model internal checkpoint/restart process. This substantially reduces pressure on the disk subsystem. Moreover, checkpointing can be performed asynchronously and several times per single job, and restarts in case of failures can be launched with minimal loss in computing time.





For efficient and parallelised checkpoint/restart, ICON creates restart information in the form of multiple files in the NetCDF format, which handles the minimum amount of information to continue a run with bit-identical results. For ICON-A, the

135 total restart output is 483 GByte, consisting of 390 output variables; ICON-O restarts consist of 83 variables with a total of 356 GBytes when using R2B9 resolution. A single checkpoint/restart is split into 82 patch files for the atmosphere and 100 patch files for the ocean, respectively, equivalent to one patch file per node. With subsequent jobs each node has to read one patch file and distribute data only within a node, provided that the number of nodes is kept fixed from job to job.

For ICON general output, numbers, frequency, and content of regular output files can be chosen freely with a namelist driven

approach. In general, I/O is performed asynchronously to the model integration on dedicated I/O nodes which allows to hide the overhead introduced by the output behind the computation (except for checkpointing). Single level global atmospheric fields in original ICON R2B9 resolution need about 78 MBytes of storage, while ocean fields can be stored in about 58 MBytes for single levels. Thus, model output can easily reach the order of Terabytes per simulation day.

All simulations performed for our MSA benchmarks run for one simulation day with 3-hourly output of 14 three-dimensional

atmospheric variables. In addition, 125 two-dimensional variables are written from the atmosphere, ocean, and land component with various output frequencies from 0.5 to 3 hours. Restart files are written at the end of the simulation. For all simulations, 17 Cluster nodes were dedicated to input/output and the total amount of storage needed for output and restart sums up to 2.2 TBytes.

## 3 Modular Supercomputing Architecture

### 3.1 Concept

The variety of application profiles running on today's HPC systems is extremely broad, as is the specificity of their requirements. Thus, serving them all with the same homogeneous hardware architecture in a suitable and energy efficient way becomes impossible. Therefore, most of the top supercomputers today are heterogeneous platforms: they combine CPUs with some kind of acceleration device, in the majority of the cases general-purpose graphic cards (GPUs). According to the June 2023 Top500

list, 185 of the 500 systems on the top500 list include GPUs (Prometeus GmbH, 2023).

Efficiently sharing resources between users in so-called *monolithic* computers, in which different kinds of hardware devices are combined within the node, is challenging. For instance, when an application running on the node is not using some of its compute elements (e.g. a GPU), making those resources available to another user is usually prevented by a critical, common node-resource (typically memory and network bandwidth) that tend to be fully utilized by the first user and which is not

manageable in a fair and consistent way.

The segregation of hardware resources, which consists of grouping equal devices within compute partitions (called *module*), makes it much easier to dynamically allocate different kinds of nodes – and therefore resources – to different users. Each application can select any number of nodes on the different modules, reserving only those resources that are needed at each point in time, and leaving the rest free for other users. The result is an optimal utilization of the overall compute resources,

which in turn maximizes the scientific output of the HPC system.



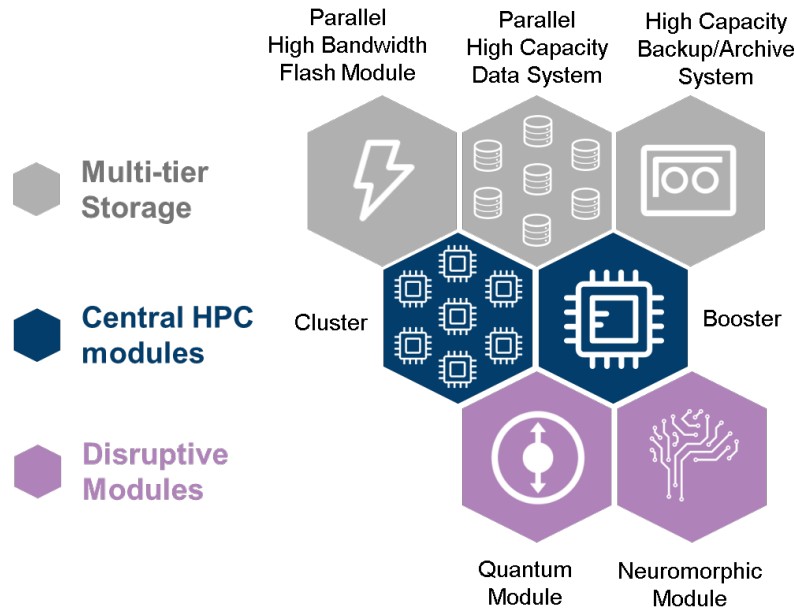

**Figure 2.** Schematic view of the Modular Supercomputing Architecture. Each module is a cluster computer of potentially large size, and its node-configuration differs from the rest of modules.

The Modular Supercomputer Architecture  (MSA, Suarez et al., 2019), leverages two or more compute modules of potentially large size – each one relatively homogeneous, and designed to best match the needs of a certain class of algorithms – to create an overall heterogeneous system (Fig. 2). For example, while the Cluster module is a CPU-based system, the processing power in the Booster module is provided by some kind of energy-efficient acceleration device (today mostly GPUs). In this manner, application parts that require high single-thread performance can use the Cluster, while the highly-scalable parts run on the Booster. The MSA concept allows for integration of additional modules to target the needs of specific user communities (e.g. data analytics or visualization modules), or to include emerging technologies such as quantum or neuromorphic computing.

With the MSA, application developers gain more freedom to choose any number and combination of CPUs, GPUs, or other accelerators. A common software stack enables them to map the intrinsic application requirements (e.g., their need for different acceleration technologies and varying memory types or capacities) onto the hardware, in order to achieve maximum performance. This is particularly interesting for workflows that are themselves somewhat modular, i.e. that perform multi-physics or multi-scale simulations (Kreuzer et al., 2021). This is frequently the case in terrestrial and atmospheric sciences, which couple different models to reproduce the highly complex interactions between the compartments of the Earth system, like atmosphere, land, and ocean, as it is also the case in ICON. Since the various modules of an Earth System Model (ESM) represent different computation behavior, they can profit from running on Cluster, Booster, or a mixture of both. In addition, the more technical issue of handling large amounts of data in model I/O may also demand for specific architectural solutions.





## 3.2 JUWELS

JUWELS (Jülich Supercomputing Centre, 2019) is an MSA system operated by JSC as a European and German computing
resource. JUWELS currently consists of two modules, a Cluster and a Booster. The Cluster – installed in 2018 – is an Atos
BullSequana X1000 system with 2567 compute nodes, equipped with Intel Skylake CPUs and Mellanox EDR InfiniBand al-
lowing for 100 Gbit/s connectivity. In 2020 it was accompanied by the Booster, a 936 node Atos BullSequana XH2000 system,
each node equipped with four NVIDIA A100 GPUs and four Mellanox HDR InfiniBand adapters (200 Gbit/s each) managed
by two AMD EPYC 7402 processors. The combined theoretical peak performance is about 80 PFLOPS. Both modules feature
an InfiniBand-based interconnect and are operated using ParTec's ParaStation Modulo software suite ParTec AG (2023).

JUWELS is catering to users needing the highest computer performance. It is used by various national and European sci-
entific communities from different domains. The 3700 GPUs of the JUWELS Booster offer accelerated tensor operations
and mixed precision, especially suited for AI research (Kesselheim et al., 2021). These are now also increasingly used by
more classical HPC applications successfully ported to GPUs, a development supported by intensive preparation and training
courses like the JUWELS Booster Early Access Program (Herten, 2021). Some recent examples of scientific applications in
the ESM community that are actively run on JUWELS Booster include deep learning methods for temperature forecasting
(Gong et al., 2022), explainable machine learning for mapping of tropospheric Ozone (Betancourt et al., 2022), atmospheric
chemical kinetics in the atmosphere-chemistry model EMAC (Christoudias et al., 2021), the Lagrangian particle dispersion
model MPTRAC (Hoffmann et al., 2022), the hydrologic model ParFlow (Hokkanen et al., 2021), the radiative transfer model
JURASSIC (Baumeister and Hoffmann, 2022) and the atmospheric component of the ICON ESM (ICON-A) (Giorgetta et al.,
2022).

## 3.3 Porting and optimization strategy

To optimally run the ICON ESM on the JUWELS system, a strategy was adopted to find a configuration in which each
component of the coupled model is matched to the best suited hardware platform. Here, the *best* fit refers to the hardware
platform and resource configuration that maximizes the performance of the application (typically defined in terms of run-time),
while simultaneously maintaining a high level of resource efficiency (considering system usage and energy efficiency).

In our experiments, the total energy consumption of all nodes involved in a particular run is used as a measure of resource
efficiency. Due to the absence of energy meters on the JUWELS nodes, it is not yet possible to measure the exact energy
consumption. As a workaround, we used a proxy for the total energy consumption defined in terms of the average total run-
time of each simulation run, multiplied by the Thermal Design Power (TDP) of all nodes involved in a particular run. The TDP
of a single node in our experiments was calculated by adding the TDP of the CPU and (if applicable), of all GPUs connected
to a node. For simplicity, we ignore the power consumed by the network or the memory in our experiments. Nevertheless,
experience shows that real-world applications never reach TDP. Thus, our proxy is rather conservative and overestimates
power consumption. The ideal representation for resource efficiency then translates to minimizing this proxy metric.



Taking both run-time and energy consumption metrics into account, it was observed that the best configuration for running the coupled model is with ICON-A running on the Booster module, and ICON-O running on the Cluster module along with the I/O servers of both components. ICON-A is naturally suited to run on a Booster module as large parts of the code-base have been optimized to make use of GPUs for faster calculations (Giorgetta et al., 2022). The ocean code has not been ported to GPUs because preliminary studies indicate that the required porting effort would be overproportionally higher than the expected performance gains. Given this constraint, the configuration that minimized energy consumption while simultaneously maintaining a high application performance was found to be through the execution of ICON-O on the Cluster module. For similar reasons, the I/O servers for both components were executed on JUWELS Cluster. The MSA experiments on JUWELS presented in Sect. 4 all rely on these basic choices, just the number of nodes have been adjusted to achieve optimal load balancing.

To demonstrate the effectiveness of the MSA for running coupled ESMs like ICON, we compare and contrast the modular approach to the standard approach of running all components of the model on a homogeneous hardware platform. In the ICON case, the homogeneous hardware platform was chosen to be the Cluster module of the JUWELS system, where all model components are able to run. JUWELS Cluster serves as an exemplary use case of a standard homogeneous HPC architecture, consisting of nearly or completely identical nodes connected by a standard high speed interconnect. This non-modular simulation also serves as a good benchmark for the ICON installation and software stack on the JUWELS system, as it uses a similar setup as the standard environment on the Levante supercomputer at DKRZ, where ICON is regularly benchmarked.

To determine a good baseline for comparing the modular setup with the standard (Cluster) setup, we first determine the optimal hardware configuration for the modular case in terms of the number of nodes involved for each component. Once the optimal model configuration for the MSA case was determined, this configuration was used as a baseline for comparison to the standard non-modular case. In order to make this fair, the number of ocean nodes was fixed at the value found best for the MSA case, while all other model parameters are kept identical between simulation runs. Then, the number of nodes used for ICON-A was increased until we reached user allocation limits (it is not possible for a single user to make use of more than a predefined threshold of 1024 nodes of the JUWELS Cluster module for a single simulation run), recording the total run-time and energy consumed in each case. To prove the scalability of the modular approach in running coupled ESM simulations with ICON, we performed a strong scaling experiment. The results of these studies are reported and examined in detail in the following section of the paper.

## 4 Results

### 4.1 Sweet spot analysis

To test the effectiveness of the MSA approach in comparison to homogeneous HPC systems, a natural question arises as to which configuration of nodes can be used as a reference baseline for comparison. Especially with heterogeneous workloads such as ICON, it is fairly common that one component of the coupled system is much faster than the other(s), due to differences in the physics and algorithms used by the different components themselves, as well as differences in the hardware on which





they run best. These differences frequently lead to situations where one component spends a significant portion of its overall runtime waiting for intermediate results from the other components, which can lead to large losses in efficiency, in terms of
250 time and energy.

Finding a *sweet spot* configuration of nodes that perfectly balances the components involved is important to maximize the overall efficiency of the simulations. It also serves the second purpose of finding a node configuration suitable as a baseline reference for comparison between the modular and non-modular approach and for strong scaling studies. The experiments performed to find the sweet spot settings are summarized in Fig. 3.

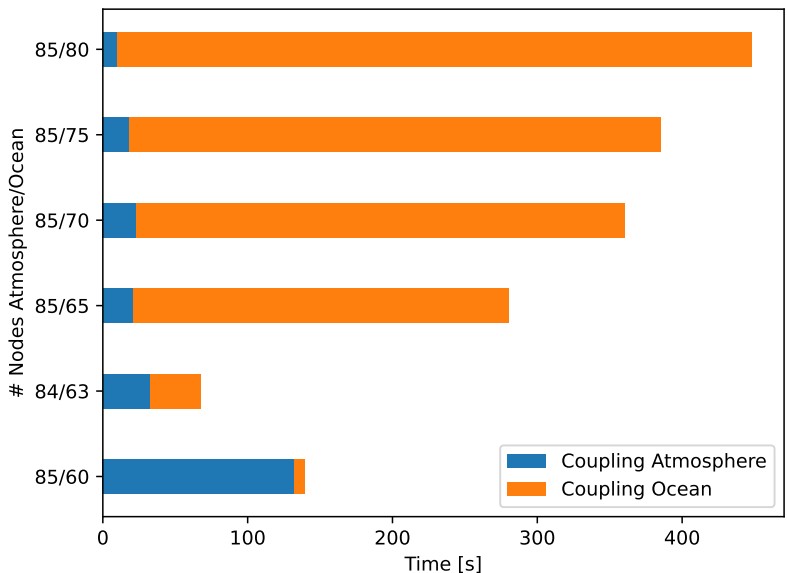

**Figure 3.** Sweet spot analysis to find a configuration of nodes that minimizes the total coupling time. On the y-axis, tuples of values represent the number of JUWELS Cluster / Booster nodes used for a simulation run. The first value of each tuple signifies the number of nodes used by ICON-A for this run, and the second value of the tuple signifies the total number of nodes used by ICON-O. The number of Cluster nodes (17) dedicated to I/O is not taken into account, since it is kept constant across all experiments.

We started out with an initial guess using a ballpark estimate of the minimum number of nodes needed for each component, considering the memory requirements of each component of the model. Keeping all other parameters fixed, we vary the number of Cluster and Booster nodes simultaneously, in order to minimize the total coupling time of the model.

The coupling time of the atmosphere component represents the cumulative time that the atmosphere component of the model spends waiting for the ocean component during a simulation run. A larger atmospheric coupling time signifies that ICON-A
260 runs faster than ICON-O and we can potentially achieve better performance by either increasing the number of ocean nodes, or decreasing the number of atmosphere nodes used for the simulation. Similarly, the coupling time of the ocean component represents the cumulative time ICON-O spends waiting for ICON-A during a simulation. The total coupling time represents the sum of the waiting times of the atmosphere and ocean components. From the optimization standpoint, in terms of both





performance and energy consumption, this time should be minimized as it is idle time that the two components spend waiting
for each other, during which no calculations are performed in the waiting component.

Based on the results of prior standalone atmosphere-only runs, the atmosphere nodes were set to 85 JUWELS Booster nodes,
using 4 GPUs and 4 cores per node, with 1 GPU per core. The number of nodes used by ICON-O were initially set to be 80,
using all available cores on a JUWELS Cluster node (48) as a starting point for the study. For the first trial node configuration
(85/80), it was observed that a significant portion of the overall run-time of ICON-O was spent waiting for intermediate results
from ICON-A. The atmosphere component also spends some time waiting for the ICON-O, but this time paled in comparison.

This information was sufficient to give clues about the ocean component being much faster than the atmosphere component
for this configuration of nodes (85/80). This means that the resources allocated to ICON-O can potentially be reduced without
loosing runtime performance. Alternatively, one could allocate more resources to ICON-A to achieve similar results. The
former approach of reducing the resources allocated to ICON-O was chosen over the latter, as this was found to to be more
resource efficient.

The analysis was repeated for other node combinations by reducing the number of JUWELS Cluster nodes by steps of 5,
until the atmosphere component became faster than the ocean component. This state is reached for the combination (85/60)
Booster/Cluster nodes. The optimal configuration of nodes that minimizes the total coupling time lies in between Booster/-
Cluster combinations (85/65) and (85/60) and was empirically determined to be (84/63) by tweaking the nodes configurations
in steps of 1. For this combination of nodes the times spend by the coupler module in ICON-A and ICON-O are almost equal
and the total coupling time is minimal. It has to be stated that another 17 Cluster nodes have been reserved as I/O servers and
their number remains constant throughout all simulations reported in this work.

### 4.2 Comparison with simulations on non-modular architectures

The optimal node configuration found through the sweet spot analysis is used as a reference to compare the MSA setup to the
non-modular case for the analysis of runtime and energy efficiency. In order to make a fair comparison, the number of ICON-O
nodes is fixed to the number that was found to be optimal for the MSA case and all other model parameters are kept identical
between simulation runs. For the homogeneous Cluster setup the number of atmosphere nodes was increased until we reached
user allocation limits (it is not possible for a single user to make use of more than a predefined threshold of 1024 nodes of the
JUWELS Cluster module on a regular basis). In both cases we recorded the total time and energy consumed by the simulations.

The total power consumed by each experiment is estimated using Thermal Design Power (TDP) as a metric for power
consumption. The TDPs of all processing elements (CPUs and GPUs) involved in the simulation runs are described in Table 1.
This information has been taken from the hardware specifications of the individual vendors (Intel, AMD and NVIDIA).

Table 2 summarizes the results of the comparison between the modular and non-modular architectures based on runtime
and energy consumption. Runtimes for ICON-A are longer than for ICON-O and determine the overall runtime. For the non-
modular setup we increased the number of nodes used for ICON-O until the workload for atmosphere and ocean is balanced,
which is the case with 780 nodes used by ICON-O, summing up to a total of 860 Cluster nodes used by the ICON simulation
including the I/O nodes. With the MSA setup, the coupling times in both components are reduced compared to the cluster-only



**Table 1.** Thermal Design Power (TDP) of individual processing elements involved in the simulation runs.

| Architecture | Chip | Thermal Design Power (W) |
|---|---|---|
| Cluster (CPU) | 2 x Intel Xeon Platinum 8168 | 2 x 205 = 410 |
| Booster (CPU) | 2 x AMD EPYC 7402 | 2 x 180 = 360 |
| Booster (GPU) | 4 x Nvidia A100 | 4 x 250 = 1000 |

simulation, most significantly in ICON-O where the reduced number of communication partners resulted in reduced network traffic. It can be observed that even though the modular and non-modular approaches are nearly identical in terms of runtime, the modular approach results in an overall reduction in energy consumption of 59 %, based on the total TDP of all CPU and GPUs involved multiplied by the total run-time of the simulation.

**Table 2.** Comparison between non-MSA and MSA architectures in terms of runtime and energy consumption. Energy consumption is computed by multiplying the total TDP of all processing elements used in the simulation multiplied by the total runtime.

| Architecture | # nodes Atm. | # nodes Ocean | # nodes I/O | # nodes total | Time Atm. [s] | Time Ocean [s] | Coupl. Atm. [s] | Coupl. Ocean [s] | Total Power Metric [W] | Total Energy [MJ] |
|---|---|---|---|---|---|---|---|---|---|---|
| Cluster | 780 | 63 | 17 | 860 | 1336 | 1275 | 55 | 154 | 352600 | 471 |
| MSA | 84 | 63 | 17 | 164 | 1321 | 1291 | 35 | 33 | 147040 | 194 |

The TDP metrics used for the computations are maximum values and assume that all nodes were being utilized at full capacity. While this might be true for non-modular runs, this is not the case with the modular approach. For all simulation runs conducted in this paper, only 4 out of the 24 CPU cores present on each JUWELS Booster node are used to run the atmosphere component. Therefore, the real savings in terms of energy consumption using the modular approach is likely to be even larger than reported here, and serves as an opportunity for model performance optimization in future work.

### 4.3 Scaling test

To further prove the efficiency of the MSA in running coupled ESM simulations with ICON, a strong scaling test was performed with the modular configuration. This was done to evaluate if the well known good scaling behavior of ICON also covers the coupling overhead introduced by MSA. To do so, the problem size was fixed at the base configuration (R2B9, 1 simulation day) and the number of nodes used by both ICON-A and ICON-O were subsequently raised for each simulation by a factor of $\sqrt{2}$ until they reach the current user allocation limits on JUWELS Booster (384 nodes). This rather small factor was chosen to accommodate enough experimental data before reaching the user allocation limits.

The results of this strong scaling test are summarized in Table 3 and Fig. 4. It is observed that the modular approach scales reasonably well up to (237 Booster / 178 Cluster) nodes and is falling off thereafter. The scaling behaviour is found to be very



similar to standalone runs of ICON-A which we performed on the JUWELS Booster system, lending credibility to the use of the modular super-computing architecture to perform ESM simulations.

In ICON, the compute domain in each MPI process is divided into blocks of a fixed length (nproma). The parameter nproma determines the length of the inner loop, and has to be specified at run-time and adjusted properly for each architecture. On
CPUs nproma is typically a small value which does not need to be changed with the number of nodes. This is set to 32 for the atmospheric component and 8 for the ocean component. In contrast on GPUs, nproma has to be set as large as possible – ideally resulting in a single local block per MPI process – in order to reduce the memory traffic between host and accelerator. With a growing number of MPI processes the local domains get smaller and thus nproma has to be reduced as well in order to allow for safely copying of and looping over non required array elements. The GPU memory on JUWELS is sufficiently large
to host the local compute domain in one single block even in our base configuration with 84 Booster nodes. This is the only parameter that is varied between the scaling simulations, keeping all other parameters of the simulation consistent across runs. Our choice of nproma for the different Booster node counts are also presented in Table 3.

From the sums of the runtimes spend in the integration loops of ICON and the coupling overhead times one can also estimate the throughput of ICON in MSA mode, which is given in Table 3 as simulation days per day (SDPD). With the
330 resource-efficient baseline simulation (84 / 63 Booster/Cluster nodes) the throughput is 76 SDPD, the highest value is reached on 335 / 251 (Booster / Cluster) nodes with 170 SDPD.

**Table 3.** Experimental data for strong scaling test of the MSA setup on the JUWELS system. Model Throughput in SDPD is calculated for the maximum of the integration times of ICON-A and ICON-O.

| # nodes Booster | # nodes Cluster | nproma | Total Time Atm. [sec] | Total Time Ocean [sec] | Int. Time Atm. | Int. Time Ocean | Atm. coupling waiting time | Ocean coupling waiting time | SDPD |
|---|---|---|---|---|---|---|---|---|---|
| 84 | 63 | 64508 | 1346 | 1312 | 1131 | 779 | 50 | 19 | 76 |
| 119 | 89 | 46156 | 1072 | 1035 | 913 | 733 | 23 | 233 | 95 |
| 168 | 126 | 32981 | 792 | 761 | 668 | 514 | 23 | 156 | 130 |
| 237 | 178 | 23628 | 646 | 600 | 531 | 399 | 36 | 168 | 163 |
| 335 | 251 | 16820 | 613 | 569 | 508 | 283 | 71 | 191 | 170 |

Figure 5 shows the scaling behavior of the total runtime composed of the model time spend in the time integration loop, the model initialization, and the writing of restart files. The initialization phase accounts for about 15 % of the total runtime in our simulations. For the climate simulations envisaged within the ICON ESM the model initialization can be neglected. Similarly,
the writing of restart files takes only a small portion of the total runtime and is performed only rarely. The regular writing of netCDF output (ICON timer *wrt_output*) is done asynchronously on dedicated Cluster nodes. Care needs to be taken that writing times are shorter than the integration times of the model time loops. If this limit is reached it may be advisable to either



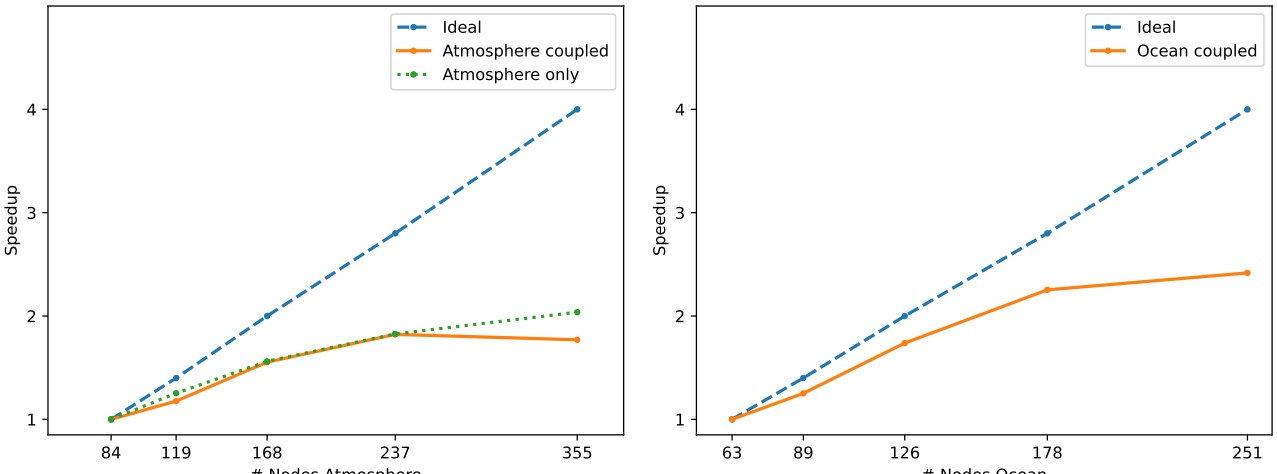

**Figure 4.** Strong scaling test using the MSA concept for the coupled components ICON-A (left) and ICON-O (right). Scaling lines (orange) are based on model integration times excluding initialization and writing of regular output. For ICON-A, scaling of the standalone atmosphere model is given for comparison (in green).

increase the number of I/O servers or to reduce output amount or frequency. For our simulations the time needed for writing output is always smaller than the model integration time.

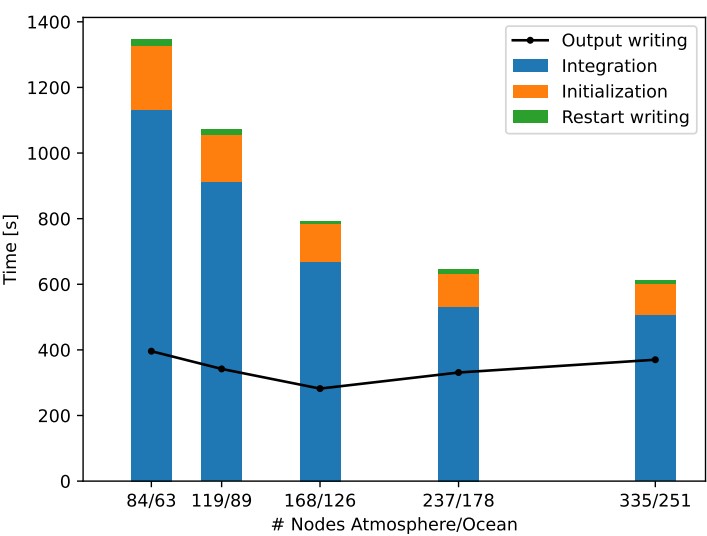

**Figure 5.** Major individual timers of ICON for the strong scaling experiments. Shown are the maxima of the timers in ICON-A and ICON-O.





## 5   Discussion

The requirements of the different components of a complex model system like ICON must be taken into account when adapting it to heterogeneous hardware computers, especially when mapping model code to different kinds of hardware. For the atmosphere-only simulations performed on JUWELS Booster previously described in Giorgetta et al. (2022), initially the output server was mixed in on the host CPUs of the GPU nodes, which introduced a significant imbalance in the node usage. This is a strong indication for network jitter where the halo exchange and the asynchronous data exchange by one-sided communication interfere. Moving the output severs to dedicated nodes led to a substantial speed-up in the wallclock time by about 15 %. However, using dedicated GPU nodes for this purpose, in which context only the host CPU is used, means letting four GPUs idling, which is a serious waste of resources. Thus, we decided here to employ CPU-only nodes of JUWELS Cluster for I/O – essentially disentangling the network resource usage and making better use of the GPUs.

As the target here is not running atmosphere-only experiments, but coupled ones where the ocean provides sufficient performance on CPUs, the selection of running this particular model component on CPU-only resources is another approach to disentangle the network traffic. As ICON-O is using a global solver for the sea level height, another different type of communication is entering the network and more jitter would be disturbing the run-time if the resources would be interleaved on the host CPU within a GPU node. In essence, currently the best way to disentangle the network usage patterns is using a hybrid setup, where the resources are used in a way as dedicated as possible. This in turn means that for our experiments the ICON-A, which is optimized for GPUs, runs on the JUWELS Booster, while dedicated CPU nodes of the JUWELS Cluster are deployed for I/O and ICON-O calculations, respectively. Efforts are now ongoing to also port ICON-O to GPU architectures, enabling GPU-only simulationd with the ICON ESM on the long run. We state here that with our setup ICON-O is always faster than ICON-A. In the (84/63) node configuration the ocean module accounts for about 17 % of the total energy consumption. Estimating similar behaviour than for ICON-A, an ocean porting would result in additional savings of 5 to 10 % for the coupled experiment. In simulations where ICON-O is run in a higher resolution than ICON-A or with a computational telescope (Hohenegger et al., 2023), more ocean nodes will be needed and energy saving potential is higher.

The ICON internal model structure is based on the multiple-program-multiple-data (MPMD) model. Therefore, the model internal structure is well prepared for distributing each of the specific components: atmosphere, ocean, and the respective output servers. From the parallel progamming perspective, the same MPI environment is running on Cluster and Booster. Therefore, one model can be compiled with the required compiler for the CPUs and the second one for the GPUs. An Application Binary Interface (ABI) compatible MPI library is used for both compiler variants, and the remaining set of required libraries is provided for both compilers. Launching this particular type of a MPMD execution model relies on the heterogeneous job support of the Slurm HPC workload manager.

In our case we had to use initially a private installed software stack based on the system provided compilers and ParaStation-MPI (here version 5.5.0-1). For initial porting and exploring new features we tend to favor such a setup, as said: compiler and MPI from system side and remaining software stack build by ourselves. System side software stacks are much easier to adapt, when an application is working and tested, and in principle favorable. Another challenge was and still is associated with the





Slurm scheduler (backfill), which does not favour the allocation of resources simultaneously on both Cluster and Booster and

can result in long queuing times. So, essentially, it can be a cumbersome and hard process to get modular application setups like the one of ICON working and requires a close collaboration with system administrators at least in the pioneering phase.

## 6   Conclusions

This work demonstrates the applicability and benefits of the modular supercomputing approach to efficiently run ESMs such as ICON on heterogeneous HPC architectures. In particular, for our test case we found that the modular approach presents

an opportunity to obtain a significant reduction in energy consumption of 59 % for running such workloads, while offering comparable levels of runtime performance to standard non-modular supercomputing architectures. It is general understanding that on climate timescales the throughput of an ESM must be greater than 100 SDPD in order to perform meaningful simulations in a reasonable time. On the JUWELS system, we are already able to excel this goal with a horizontal resolution of 5 km. Outside the context of this project, ICON-A is running in resolution R2B10 (2.5 km) on JUWELS Booster, occupying a

significant portion of its GPU resources. In order to achieve effective ICON ESM simulations in this or even in R2B11 (1.25 km) resolution, HPC systems in the exaflop range will be needed. EuroHPC and Research Center Jülich will provide such a system – JUPITER – based on the MSA concept by late 2024, and ICON is preparing for the first kilometer-scale coupled climate simulations there.

Through interactions with the developers of large scientific codes such as ICON, it was noted that the porting of scientific

code tends to take place in a phased fashion, wherein a subset of the code that is most suited to take advantage of a new hardware platform is ported first, before attempting to port other parts of the source code. This approach is transferable to other multi-physics applications as well. In traditional architectures, one might have to *waste* hardware resources during development and testing, while migrating subsets of a large application such as ICON to a new hardware platform such as a GPU. The modular approach provides developers with more freedom to pick and choose nodes with differing hardware configurations to run

subsets of the code during development and testing, which can lead to higher developer productivity and resource efficiency. However, an imported pre-condition to the successful application of the modular approach is that it requires the design of the code itself and by extension, the physical problem that one is trying to simulate to be modular in nature. Whenever there is a clear separation of concerns, for example: in multi-physics simulations such as the ones performed within ICON, the use of modular architectures is natural and expected to provide significant benefits. It must be noted though that the modular

architecture is not a silver bullet that is expected to be beneficial to all types of problems.

Hallo Olaf, wir testen jetzt....

In recent years, the awareness of responsible energy use has steadily increased. Supercomputing centers are known to be large energy consumers and there is growing pressure from the funding organizations to keep energy costs at a low level. We thus believe that the Modular Supercomputing Architecture can play a leading role in running modular workflows like large

scale Earth System Models efficiently on HPC systems in the near future.



*Code availability.* Simulations were done with the ICON branch 2.6.6-rc. This source code can be downloaded from the publically available repository https://doi.org/10.17617/3.4NHKPH (last access: 14 August 2023). The ICON model is distributed under an institutional license issued by DWD as well as under a personal non-commercial research license distributed by MPI-M (https://code.mpimet.mpg.de/projects/iconpublic/wiki/How_to_obtain_the_model_code, last access: 14 August 2023). By downloading the ICON source code, the user accepts the
410 license agreement.

*Author contributions.* AB prepared and optimized the ICON model for usage with the MSA on the JUWELS system and conducted most of the experiments. OS and RR supported the experiments and contributed with analysis and discussion of the results. CM designed the experiment setup, analysed the model results and prepared Figs. 3 to 5. LK, HH, and DK are developers of the ICON model and provided expertise in high-resolution modeling and the coupled setup for the MSA. LK was responsible for downloading and preparing input data sets
for ICON. NE and ES developed the concept for this study and steered the experiments to study the specifics of the MSA concept on the JUWELS system. AB and OS wrote the manuscript with contributions from all co-authors.

*Competing interests.* The authors declare that no competing interests are present.

*Acknowledgements.* This work was supported by the Helmholtz Association of German Research Centres (HGF) through the Joint Lab Exascale Earth System Modelling (JL-ExaESM), as well as by the AIDAS project of the Forschungszentrum Jülich and CEA. The authors
gratefully acknowledge the computing time and storage resources granted through the ESM partition on the supercomputer JUWELS at the Jülich Supercomputing Centre, Forschungszentrum Jülich, Germany. We are thankful to Dmitry Alexeev, Nvidia, and Will Sawyer, CSCS, PRACE project pra127, and the application support and systems operations teams at JSC.



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
