# Peer review of "Earth system modeling on Modular Supercomputing Architectures: coupled atmosphere-ocean simulations with ICON 2.6.6-rc"

_EGUsphere, 2023_

## Author Comment (AC2)

**Reply to reviewer comment RC1**

*The study of Bishnoi et al. shows on the one hand that it is possible to run ICON on a heterogeneous architecture, with the atmosphere part running entirely on GPUs and the ocean part and the input and output on dedicated CPUs nodes, and on the other hand that considering the energy consumption, the performance of ICON on the heterogeneous architecture is improved by 59% compared to a pure CPU based architecture. The study is definitely very relevant for the atmospheric model community, since the ICON model is actively used by a large number of institutes. Because of the ever increasing share of boosters in the new supercomputers, it is necessary that at least large parts of ICON can run on boosters (e.g. GPUs). This is especially important with respect to the already existing and future exascale computers. The study therefore shows the feasibility. The fact that the performance is additionally increased, due to a lower energy consumption, can be considered as success and is quite reasonable to be presented in a publication. Furthermore, the study also show that the current architecture of the JUWELS system is definitely useful. Therefore, I can definitely recommend to publish the presented study and think that it is absolutely suitable for GMD.*

We thank the reviewer for the valuable and encouraging remarks, which led to further improvement of the paper manuscript. In the following we tried to carefully address your comments and propositions. In addition, we added a short paragraph summarizing the discussion that arose from community comment CC1.

During the preprint phase, we detected an error in our Thermal Design Power (TDP) calculation. In fact, the TDP reported for JUWELS Booster was 400 W per CPU instead of 250 W. Taking this into account, our estimates for energy saving by using MSA instead of JUWELS Cluster reduced from 59% to 45%. We are convinced that also this somewhat smaller improvement justifies publication and hope that the rough estimate can be replaced by more accurate measurements in the future.

*I have only one general remarks:*

*Since ICON is a scientific test case, I would find it useful to present at least a few scientific results in a short subsection, regarding whether they are identical or almost identical, regardless of whether ICON runs on a homogeneous (cluster) architecture or on a heterogeneous architecture (cluster/booster). Perhaps a monthly average of the temperature or the zonal wind could be presented here.*

In this work we have intentionally omitted any comparison of meteorological fields between simulations performed on different supercomputing architectures. On the one hand, our simulations are too short to analyze such model results and its variations, as the development of most meteorological variables after 24 hours of forecast will largely depend on the choice of initial conditions. On the other hand, ICON is known to provide non-bit-identical results, even for different node numbers using the same architecture. We compared our simulation results after 1 simulation day with JUWELS Booster node variations from 84 to 335 nodes and found differences in mean and standard deviation of less than 0.001 % for 2-meter temperature and surface pressure between the simulations. For 10-meter zonal wind these differences are < 0,3% (mean) and < 0,04% (standard deviation). These results are well in line with previous ICON experiments performed for benchmarking on JUWELS and in the HD(CP)² project (https://www.dkrz.de/de/projekte-und-partner/HLRE-Projekte/focus/hdcp2).

*And some minor remarks:*

*Line 13/14: "… was found for the combination 84 GPU nodes on the JUWELS Booster module and 80 CPU nodes on the JUWELS Cluster module …" --> "… was found for the combination 84 GPU nodes on*

*the JUWELS Booster module to simulate the atmosphere and 80 CPU nodes on the JUWELS Cluster module …"*

done

*Line 42: "by a factor of 1 million" --> "by a factor of more than 1 million"*

done

*Line 57/58: I would also integrate here the acronym DKRZ, C2SM, and KIT*

done

*Line 61/62: "the performance of the ocean component on CPUs is still satisfactory". Does it make sense to say it that way? Shouldn't one rather write "it is not yet possible to simulate the ocean on GPUs"? Later you write that there is a project for it.*

We agree that it is desirable to also port ICON-O to GPUs. In fact, efforts to do this are already advanced. We changed the sentence accordingly.

*Line 78: "Jülich Wizard for European Leadership Science (JUWELS) Jülich Supercomputing Centre (2019)" --> "Jülich Wizard for European Leadership Science (JUWELS, Jülich Supercomputing Centre (2019)"*

done

*Line 83-89: I would suggest: "In Sect. 2 we provide a comprehensive description of the ICON model and its specific setup. Sect. 3 presents a brief overview of the MSA, starting with an introduction to the concept (Sect. 3.1), the presentation of the modular hardware and software architecture of the JUWELS system at JSC (Sect. 3.2), and the strategy for porting the ICON model to the MSA, with a detailed explanation of the rationale behind each decision we made (Sect. 3.3). In Sect. 4 results from our analyses for finding a sweet spot configuration for ICON, the comparison to a non-modular setup, and strong scaling tests are provided. In Sect. 5 specific challenges and considerations associated with porting such complex codes as ICON to the MSA are discussed, and in Sect. 6 the summary and conclusions of this study is provided."*

Lines 83-89 (now lines XXX-XXX) have been rewritten according to the suggestions of both reviewers.

*Line 95: "of this paper". I would rather write (also in all other cases further down in the text) "of this publication" or "of this study".*

done

*Line 96: I would also mention R2B09 here.*

done

*Line 96: "Thus, the grids" -> "Thus, the horizontal grids"*

done

*Line 117: The start date of the ICON simulation is 20 January 2020. What is the end date?*

The end date is 21 January 2020, 0 UTC, resulting in simulations of 24 hours duration. We changed the first sentence in Sect. 2.1 accordingly.

*Line 186: How many CPUs and how many cores has one cluster node?*

We added the number of CPUs (2) and cores per CPU (24) for JUWELS Cluster in line 186 (now line XXX).

*Line 197: "Ozone" --> "ozone"*

done

*Line 200: "ICON ESM" --> "ICON-ESM"*

done

*Line 217: I would delete "naturally"*

done

*Line 226: "… homogeneous hardware platform." --> "… homogeneous hardware platform, using only CPUs."*

done

*Line 229: "nodes" --> "CPU nodes"*

done

*Line 225-240: I would not speak from a modular approach or case, this is in my opinion confusing, because ICON has also modules. I would suggest to change modular to "heterogeneous" and non-modular and standard to "homogeneous". That would also be relevant in the rest of the text (Sect. 4 and 5).*

We refer here to the modules of the supercomputing hardware resources, a central concept of MSA. Unfortunately, the modular approach in ICON can give rise to misunderstandings about what "modular" means. We changed "modular approach" to "modular hardware approach" in line 225 and decided to keep "modular" otherwise.

*Line 233: "... for each component." --> "for each model component (atmosphere, ocean)."*

In addition to atmosphere and ocean, the I/O infrastructure must be considered as an ICON component, too. We added the actual three components in line 228 (now line XXX).

*Line 267: "4 GPUs and 4 cores per node, with 1 GPU per core." Is this correct? I thought there are 2 CPUs per node, i.e. I think 48 cores, or?*

Here you mix up Booster and Cluster nodes. The Booster nodes have two Intel Xeon Gold 6148 CPUs with 20 cores each, hosting four NVIDIA A100 GPUs. Thus we use 4 CPU cores and 4 GPUs per node. For clarification we changed "cores" to "CPU cores".

*Line 268: "(48)" --> "(48 cores/node)"*

done

*Line 269: "(85/80)" --> "(85 Booster nodes/80 Cluster nodes)"*

done

*Figure 3: Please change "JUWELS Cluster/ Booster nodes" to "JUWELS Booster / Cluster nodes" in the caption*

done

*Line 286: "MSA case and …" --> "MSA case (63 nodes) and …"*

done

*Line 289: "both" --> "all"*

done

*Line 294: "Runtimes for ICON-A are longer than for ICON-O and determine the overall runtime" --> "With 85 nodes for ICON-A and 63 nodes for ICON-O, runtimes for ICON-A are much longer than for ICON-O and determine the overall runtime"*

For both cases shown in Table 2, runtimes for ICON-A are slightly longer than for ICON-O. This is also true for all MSA simulations performed in our study. Thus we decided to keep the sentence but we added "in both configurations" for clarification.

*Line 295: I would delete "For the non-modular setup"*

Here we describe how we derived the number of nodes used for our comparison in the non-modular, GPU-only configuration. As those model simulations (increasing the number of CPU nodes for ICON-A) were not part of the actual study, we added "based on previous experience".

*Line 295: "ICON-O" --> "ICON-A"*

done

*Line 296: "ICON-O" --> "ICON-A"*

done

*Line 301: How many SDPDs are simulated with 780/63 nodes? Maybe you can still integrate this value in Table 2.*

The throughput of the non-MSA simulation is similar to that of the MSA simulation using 84 Booster nodes and 80 cluster nodes, as can be seen from the similar total runtimes. Unfortunately, the model timers for initialization and restart writing have not been recorded for the non-MSA simulation (which was also done previously to this work as noted above). For reasons of resource efficiency and core-hours on JUWELS Cluster available to us, we decided not to restart this rather large simulation.

*Table 2: I would find it useful to include not only the final result (780/63) in the table, but also the results of (84/63) and the steps in between.*

As mentioned above, these steps, purely on the JUWELS Cluster, have not been performed within this study.

*Line 334: What is the reason for this (to use only 1 core/GPU)?*

Sorry, but we can't find the location of the text you are referring to.

*Line 317: I would insert "(see Fig. 4)" at the end of the sentence.*

This is done at the beginning of the next sentence.

*Line 332: "Figure 5" --> "Fig. 5"*

According to GMD submission guidelines, the abbreviation "Fig." should be used when it appears in running text and should be followed by a number unless it comes at the beginning of a sentence.

*Line 332: Why don´t you increase the number of I/O nodes during your scaling test?*

As I/O is done asynchronously during ICON model integration, the number of I/O nodes needs to be chosen such that I/O is performing faster than the model integration time. As you can see from Fig. 5, this is always the case, even with our largest node configuration. If we decided to scale up even further or to increase the resolution, we'd also need to increase the number of I/O nodes.

*Fig.4 (left): The orange line shows a decrease in the speedup from 237 nodes to 355 nodes, but in table 2 there is still a decrease in the Int. time from the atmospheric compound, so there should be at least a small increase in the speedup.*

Thank you for pointing to this inconsistency, which has also been noticed by reviewer 2. The numbers given in Table 3 are correct, but we used erroneous values for producing Fig. 4. Indeed, speedup for 355 nodes is well above two and speedup still increases up to the maximum node count. Figure 4 has been corrected in the final version of the manuscript.

*Line 379: "In particular, for our test case we found that …" --> "In particular, for our test case, a coupled ICON simulation, we found that …"*

done

*Line 384: "… ICON-A is running …" --> "… ICON-A is already running …"*

done

*Line 401: Please delete "Hallo Olaf, wir testen jetzt.…"*

done

---

## Author Comment (AC3)

**Reply to reviewer comment RC2**

*The paper by Bishnoi et al. describes the work on using the modular supercomputing architecture (MSA) for coupled atmosphere and ocean simulations of the ICON model. The authors find that the MSA-approach improves the energy consumption by 59% compared to running the entire coupled model on the CPU nodes of the JUWELS Cluster supercomputer. The paper is well written and the results are very relevant for the audience of GMD. It is overall well structured and provides all the necessary details to understand how the results were obtained. There are a few places which I describe in detail in the specific comments below where the text is difficult to understand and should be improved. I recommend to publish the paper once these comments are addressed.*

We thank reviewer 2 for the constructive comments and suggestions and for the positive impression our work has made. In the following we tried to carefully address your issues and hope that our corrections will result in a further improved paper manuscript. In addition, we added a short paragraph summarizing the discussion that arose from community comment CC1.

During the preprint phase, we detected an error in our Thermal Design Power (TDP) calculation. In fact, the TDP reported for JUWELS Booster was 400 W per CPU instead of 250 W. Taking this into account, our estimates for energy saving by using MSA instead of JUWELS Cluster reduced from 59% to 45%. We are convinced that also this somewhat smaller improvement justifies publication and hope that the rough estimate can be replaced by more accurate measurements in the future.

*Specific comments:*

*line 33-35: This sentence is difficult to understand since it is too long. It should be split in two: "The code of more advanced complex climate models is composed of many different kinds of operations, e.g., ... Another level of complexity is added through optimisations for specific computing hardware (Lawrence et al., 2018)."*

done

*line 62: up to 10% overall speedup according to line 360 doesn't sound satisfactory*

In line 360, we refer to potential additional energy savings when porting ICON-O to GPUs, not speedup. We changed "savings" to "energy savings" in line 360 (now line 358) for clarification. Work on GPU porting of ICON-O is ongoing, aiming at better standalone performance and, in coupled mode, reducing parts of the coupling overhead (data transfers CPU host – GPU device).

*line 83-87: Mixing sections and subsections is confusing. Better write something like: "In Section 2 we provide a comprehensive description of the ICON model and its specific setup. Section 3 presents a brief overview of the MSA, with an introduction to the concept (Section 3.1), a description of the modular hardware and software architecture of the JUWELS system at JSC (Section 3.2) and the strategy for porting the ICON model to the MSA (Section 3.3). Section 4 contains the results from our analyses for finding a sweet spot configuration for ICON (Section 4.1), the comparison to a non-modular setup (Section 4.2), and strong scaling tests (Section 4.3)."*

Lines 83-89 have been rewritten according to the suggestions of both reviewers.

*line 225-241: The goal of these two paragraphs should be made clearer. In particular the formulation "until we reached user allocation limits" in line 237 sounds to me as if ICON-A was still slower compared to ICON-O and more nodes beyond the allocation limit would have improved the energy efficiency of the homogeneous configuraton further. This is not the case according to line 295*

*which states that ICON-O and ICON-A are balanced. Also it should be mentioned that this will be described in much more detail in Section 4. I suggest to replace the entire two paragraphs (line 225-241) with something like: "To quantify the benefit of the MSA approach we compare the energy consumption of an optimal MSA configuration with a homogeneous setting in which the entire coupled model is run on the same type of nodes while keeping the run-time roughly the same. Since not all model components of ICON can take advantage of GPUs we use the CPU nodes of the JUWELS Cluster module as a baseline for this comparison. The run-time is kept roughly the same by using the same number of nodes for the ocean component. Both configurations (MSA and homogeneous baseline) are optimized by adjusting the number of nodes used for the atmosphere component such that waiting times between model components are minimized. All other model parameters are kept the same. The process of finding the optimal configuration is described in detail in Section 4.1 for the MSA configuration and in Section 4.2 for the homogeneous baseline. In addition, we performed a strong scaling experiment to prove the scalability of the MSA approach which is presented in Section 4.3."*

The last paragraph of Section 3.3. has been re-arranged according to your suggestions.

*Figure 3: The absolute values of the coupling time alone are not very meaningful in this context. Either mention already here how these times compare to the overall run-times or use percentage of the overall run-time of the simulation for the horizontal axis. This would allow the reader to immediately understand the severeness of these coupling times and it would still convey the message which configuration is best.*

We took up the reviewer's valuable suggestion and changed Fig. 3, showing now the percentage of runtime instead of absolute timings.

*line 269: If you don't show the percentage of the overall run-time in Figure 3 you should mention here the overall run-time to allow the reader to understand why it is a significant portion of the overall run-time.*

done with Fig. 3

*line 285: I love the way how you compare the MSA and non-MSA approach. I agree that it is a fair comparison. I just miss a clear statement how you chose to compare the two approaches. In my opinion keeping the number of ICON-O nodes the same is rather a matter of how you chose to compare the approaches than a matter of making the comparison fair. In principle one could also choose to compare how many SDPD the same amount of energy can achieve in which case one wouldn't keep the number of ICON-O nodes the same. I think you should replace "In order to make a fair comparison" with something like "In order to keep the run-time roughly the same"*

We agree that a comparison could also be done in different ways. We changed "In order to make a fair comparison …" to "We decided to scale up the non-modular simulation such that total run times are in the same order than for the MSA setup. To achieve this, …" for clarification.

*line 295: What exactly does it mean that atmosphere and ocean are balanced? Does it mean that coupling times are minimized or that the integration times are the same?*

We increased the number of GPU nodes used for ICON-A such that integration times for atmosphere and ocean are roughly the same. We changed "workload for atmosphere and ocean" to "workload for the integration of atmosphere and ocean" for clarification.

*Table 3: Please explain in the caption or in the main text whether the waiting time is already included in the total time or not.*

The total timers includes the waiting times both for the halo exchange and the waiting for the exchange of coupling fields. We added an explanatory sentence in the caption of Table 3.

*Figure 4: What is the relation between the numbers in Table 3 and the results shown in Figure 4? The figure shows a speedup of about 1.8x for atmosphere coupled at 355 nodes. All the speedups for the atmosphere component in Table 3 give me a speedup of at least 2 times even if I assume that the waiting time is not included in the total time. How does this fit? Why is the speedup at 355 nodes for the atmosphere lower than at 237 nodes? This is not visible in Table 3.*

Thank you for pointing to this inconsistency, which has also been noticed by reviewer 1. The numbers given in Table 3 are correct, but we used erroneous values for producing Fig. 4. Indeed, speedup for 355 nodes is well above two and speedup increases up to this node count. Figure 4 has been corrected in the final version of the manuscript.

*Minor comments:*

*line 13: "combination 84 GPU nodes" => "combination of 84 GPU nodes"*

done

*line 42: "factor of 1 million" => "factor of more than 1 million"*

done

*line 91: "non-hydrostatic atmosphere" => "non-hydrostatic atmosphere model"*

done

*line 96: It would be good to introduce the R2B9 grid already at this point which is often used later.*

done

*line 117: It would be helpful to mention already here that all experiments run for 1 simulation day.*

done

*second line of the caption of Figure 3: "Cluster / Booster" => "Booster / Cluster"*

done

*third and fourth line of the caption of Figure 3: "The number of Cluster nodes (17) dedicated to I/O is not taken into account, since it is kept constant across all experiments." => "The 17 Cluster nodes dedicated to I/O are not taken into account, since the number of IO nodes is kept constant across all experiments."*

done

*line 268: "(48)" => "(48 cores/node)"*

done

*line 269: "(85/80)" => "85 Booster nodes / 80 Cluster nodes"*

done

*line 295: "ICON-O" => "ICON-A"*

done

*line 296: "ICON-O" => "ICON-A"*

done

*caption of Table 2: Shouldn't this be rather "MSA configuration" than "MSA architecture"?*

done

*line 358: "simulationd" => "simulations"?*

done

*line 401: please remove this line "Hallo Olaf, wir testen jetzt...."*

done